# $3.5 \times 3.5\,\mu m^2$ GaN blue micro-light-emitting diodes with negligible sidewall surface nonradiative recombination

Xuelun Wang [1,2,3] ✉, Xixi Zhao [2], Tokio Takahashi[2], Daisuke Ohori [4] & Seiji Samukawa [4,5] ✉

Micro-light-emitting diode displays are generating considerable interest as a promising technology for augmented-reality glasses. However, the fabrication of highly efficient and ultra-small ( <3 μm) micro-light-emitting diodes, which are required for augmented-reality applications, remains a major technical challenge due to the presence of strong sidewall nonradiative recombination. In this study, we demonstrate a $3.5 \times 3.5\,\mu m^2$ blue GaN micro-light-emitting diode with negligible sidewall nonradiative recombination compared with bulk nonradiative recombination. We achieve this by using an ultralow-damage dry etching technique, known as neutral beam etching, to create the micro-light-emitting diode mesa. Our $3.5 \times 3.5\,\mu m^2$ micro-light-emitting diode exhibits a low decrease in external quantum efficiency of only 26% at a current density of $0.01\,A/cm^2$, compared with the maximum external quantum efficiency that is reached at the current density of $\sim 3\,A/cm^2$. Our findings represent a significant step towards realizing micro-light-emitting diode displays for augmented-reality glasses.

Microdisplays using semiconductor micro-light-emitting diodes (micro-LEDs) as light emitters are considered to be ideal candidates for next-generation virtual reality (VR) and augmented reality (AR) smart glasses that require high-resolution, high-luminance, and energy-efficient displays[1]. However, the near-eye operating environment of VR/AR microdisplays imposes strict limitations on the size and power consumption of the micro-LEDs. For AR displays, a resolution of at least 4000 pixels per inch (ppi) is required to achieve a high-quality display, which corresponds to a pixel pitch of approximately 6 μm[2–4]. To achieve this resolution, the maximum size of the micro-LEDs for each primary color should be reduced to below 3 μm. Heat generation during the operation of AR microdisplays is not allowed because the display is located very close to the human eye. Unfortunately, approximately only 50% of the injected electrical power is converted into light, even in state-of-the-art red, green, and blue semiconductor LEDs[5]. The remaining 50% of electrical power is converted into heat, which increases the temperature of the LED package without an efficient thermal management design[6]. However, it would be impractical to implement an active thermal dissipation design in AR glasses owing to the light-weight requirement. The most effective way to suppress heat generation is to reduce the operation current density of micro-LEDs. A simple estimation of a 4000-ppi microdisplay using GaN micro-LEDs suggests that the temperature increase can be suppressed to approximately 20 °C by reducing the operating current density to the order of $1\,A/cm^2$ (Supplementary Fig. 2).

The fabrication of highly efficient sub-3-μm micro-LEDs in the $<1\,A/cm^2$ current density region is a significant challenge because of

[1]GaN Advanced Device Open Innovation Laboratory, National Institute of Advanced Industrial Science and Technology (AIST), Furo-cho, Chikusa-ku, Nagoya, Japan. [2]Research Institute for Advanced Electronics and Photonics, National Institute of Advanced Industrial Science and Technology (AIST), Tsukuba, Japan. [3]Institute of Materials and Systems for Sustainability, Nagoya University, Furo-cho, Chikusa-ku, Nagoya, Japan. [4]Institute of Fluid Science, Tohoku University, Aoba-ku, Sendai, Japan. [5]Institute of Communications Engineering, College of Electrical and Computer Engineering, National Yang Ming Chiao Tung University, Hsinchu, Taiwan. ✉e-mail: xl.wang@aist.go.jp; seiji.samukawa.e2@nycu.edu.tw

the presence of strong Shockley-Read-Hall nonradiative recombination arising from mesa sidewall defects. Typically, GaN micro-LEDs are produced by inductively coupled plasma (ICP) etching of a planar LED wafer, which introduces high-density defects acting as nonradiative recombination centers to the sidewall surfaces of the micro-LED mesa owing to ion bombardment and deep ultraviolet photon irradiation[7,8]. As a result, strong sidewall surface nonradiative recombination occurs, reducing the emission efficiency of micro-LEDs by decreasing the chip size[9,10], especially in sub-3-μm micro-LEDs where the micro-LED dimension becomes comparable to the diffusion length of minority carriers in GaN[11,12]. This efficiency reduction was particularly significant in the current density region below 1 A/cm², since the external quantum efficiency (EQE) of a high-quality InGaN/GaN LED typically shows a maximum at a current density of a few to tens of A/cm² and decreases with decreasing current density owing to enhanced Shockley-Read-Hall nonradiative recombination[13].

Intrinsic surface defects such as dangling bonds in the outermost surface layer also play an important role in sidewall surface non-radiative recombination of GaN micro-LEDs. These defects act as nonradiative recombination centers and cause pinning of the Fermi level at the sidewall surface[14,15], leading to the bending of the surface band energy near the surface area and the accumulation of carriers at the sidewall surface. This accumulation provides an additional channel for nonradiative surface recombination. Jiang et al. conducted a the-oretical investigation into the impact of intrinsic surface states on GaN blue micro-LEDs grown on a *c*-plane sapphire substrate[16]. They found that the presence of intrinsic surface states reduced the maximum internal quantum efficiency (IQE) from 58% to 24% when the chip size was decreased from 300 to 3 μm, even without plasma-etching-induced surface damage.

Despite the considerable efforts to reduce sidewall surface nonradiative recombination in GaN micro-LEDs smaller than 10 μm, this problem persists. To address this issue, Wang et al. used neutral beam etching (NBE), an ultralow-damage dry etching technique for semiconductor materials[17], to fabricate GaN blue micro-LEDs[18,19]. Their 6 × 6 μm² GaN micro-LED displayed EQE vs. current density characteristics similar to those of a 40 × 40 μm² device in the current density region higher than 1 A/cm², indicating a significant reduction in dry-etching-induced nonradiative recombination. Ley et al. reported the first 2 μm InGaN/GaN blue micro-LED fabricated by ICP etching, where potassium hydroxide (KOH) treatment of mesa side-walls was employed to remove plasma-induced defects, and the sidewall surface was further passivated by a thin Al₂O₃ layer depos-ited by atomic layer deposition[20]. However, the EQE of the fabricated micro-LED decreased to approximately 17% of its peak EQE at a cur-rent density of 15 A/cm² when the current density was decreased to 0.1 A/cm².

In this work, we successfully fabricate a 3.5 × 3.5 μm² GaN blue micro-LED with negligible mesa sidewall nonradiative recombination compared with bulk nonradiative recombination of the epitaxial wafer, using the NBE process. Our devices exhibit a low decrease in EQE (approximately 26%) at a current density of 0.01 A/cm² compared to the peak EQE observed around 3 A/cm². Analysis of the EQE char-acteristics and measurement of the surface potential of the NBE-etched sidewall surface using Kelvin force microscopy (KFM) reveals that our devices effectively suppress nonradiative recombination related to not only sidewall defects generated during mesa etching but also intrinsic surface states.

## Results
### Fabrication of the 3.5 × 3.5 μm² micro-LED
The NBE system employs a unique carbon electrode featuring a high-aspect-ratio aperture positioned between the ICP discharge chamber and the etching chamber. As accelerated ions pass through the aper-ture, charge exchange with the carbon electrode efficiently neutralizes

them, resulting in a neutral beam with controlled kinetic energy directed towards the sample surface for etching[17,19]. Moreover, the carbon aperture blocks deep ultraviolet photons, allowing for ultralow damage etching of semiconductor materials. Overall, the NBE process is an effective means of providing semiconductor material etching with minimal damage.

Commercial InGaN/GaN blue LED wafers grown on patterned (0001) *c*-plane sapphire substrates using metal-organic vapor phase epitaxy (MOVPE) were utilized as epitaxial materials. The active region comprised 16 pairs of InGaN/GaN multiple quantum wells (MQWs), emitting light at approximately 458 nm. Figure 1a illustrates the schematic of the micro-LEDs fabricated in this study. Square-shaped micro-LED mesas with sizes ranging from 3 to 20 μm were etched by NBE, using SiO₂ deposited by plasma-enhanced chemical vapor deposition (PECVD) as a mask. In addition, a series of reference samples were fabricated using the conventional ICP process, fol-lowed by a KOH solution treatment to remove the plasma-induced damage layer. Subsequently, a 150-nm-thick SiO₂ film was deposited on the sample surface by PECVD as an electrical isolation and surface passivation layer. Next, a self-aligned Ni/Au p-type Ohmic contact was prepared on the mesa top (Supplementary Fig. 3), and a Cr/Au n-type contact was formed on the n-GaN surface (Supplementary Fig. 3). The thickness of the entire layer was adjusted such that the surface of the n-contact was at the same height as the p-contact. The device process was completed by the deposition of an Au/Sn multi-layer on both the p-type and n-type Ohmic contacts, which acted as a eutectic bonding layer. Figure 1b, c depict 45° tilted scanning elec-tron microscopy (SEM) images of a 3-μm micro-LED fabricated by the NBE process and the same device where the SiO₂ passivation layer was removed by wet etching using buffered hydrofluoric acid (BHF) to expose the details of the micro-LED mesa, respectively. The micro-LED mesa comprised four vertical sidewall surfaces with a very smooth surface morphology. These surfaces are close to the non-polar *m*- and *a*-planes of the GaN crystal based on the crystalline orientation of the epitaxial wafer, but they may not represent the exact *m*- and *a*-planes of GaN because of the limited precision of device processes. To avoid misleading, we refer to these surfaces as "*m*-plane-like surface" or "*a*-plane-like surface" hereafter. The depth of the mesa is approximately 700 nm. The presence of vertical nonpolar-like sidewall surfaces indicates that the chemical reaction is the dominant mechanism in the NBE etching of GaN. The mesa size was approximately 3.5 × 3.5 μm², slightly larger than the designed size of 3 × 3 μm². The size of the Au/Sn multilayer and Ni/Au p-contact (not clearly revealed in Fig. 2c) was 2 × 2 μm². Hereafter, we refer to the device size by the actual size measured by SEM observation, e.g., a 3.5 × 3.5 μm² micro-LED.

After completing the device fabrication process, the micro-LED chip was diced into a 1 × 1 mm² size and bonded to an Si submount with an electrical injection circuit using the flip-chip eutectic bonding technique (Supplementary Fig. 4). The light emission was extracted from the sapphire substrate side and measured using a Si photodiode calibrated to detect emission with a limit of 0.5 nW (Thorlabs, S130VC). The Si photodiode was placed approximately 4 mm from the micro-LED chip, which corresponds to a collection half angle of approxi-mately 52°. The light emission under high current densities was also measured using a 2-inch integrating sphere (Thorlabs, S142C) with a detection limit of 1 μW to evaluate the absolute emission effi-ciency value.

### EQE analysis as a function of current density
Figure 2 depicts the EQE measured as a function of current density for four micro-LEDs with sizes ranging from 3.5 × 3.5 μm² to 20.5 × 20.5 μm², which were fabricated using the NBE process. For comparison, a 3 × 3 μm² sample etched using ICP and treated with a KOH solution is also shown. The EQE was calculated using the

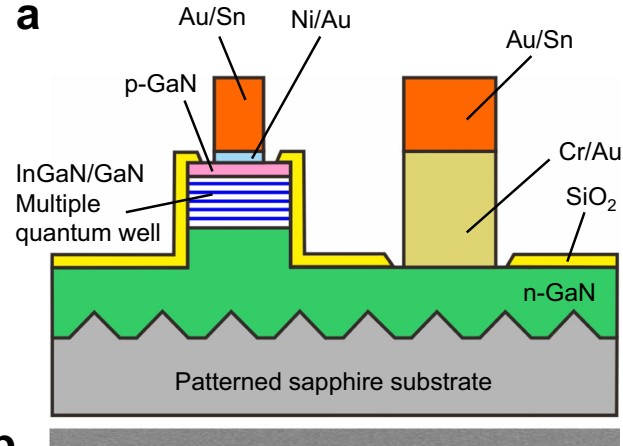

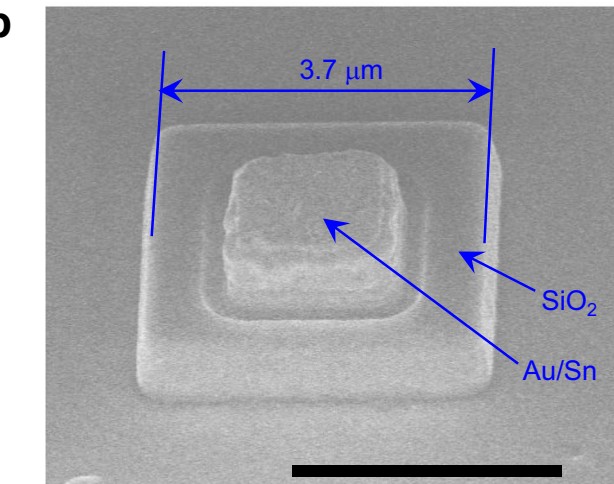

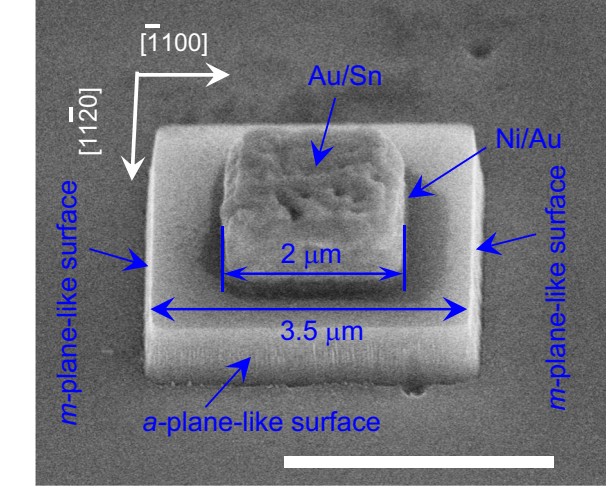

**Fig. 1 | Micro-LED structure. a** Cross-sectional schematic illustration of the micro-LEDs fabricated in this work. The sapphire substrate was thinned from the backside to 200 μm and polished to a mirror surface. **b** A 45° tilted SEM image of a 3.5 × 3.5 μm² micro-LED fabricated by the NBE process is shown. Scale bar, 3 μm. **c** A 45° tilted SEM image of the same micro-LED after the SiO₂ passivation layer was etched off by BHF solution. Scale bar, 3 μm.

following equation:

$$EQE = \frac{P\lambda e}{Ihc} \tag{1}$$

where $P$ is the measured light output power, $\lambda$ is the peak light emission wavelength, $I$ is the injected current, $c$ is the speed of light in a

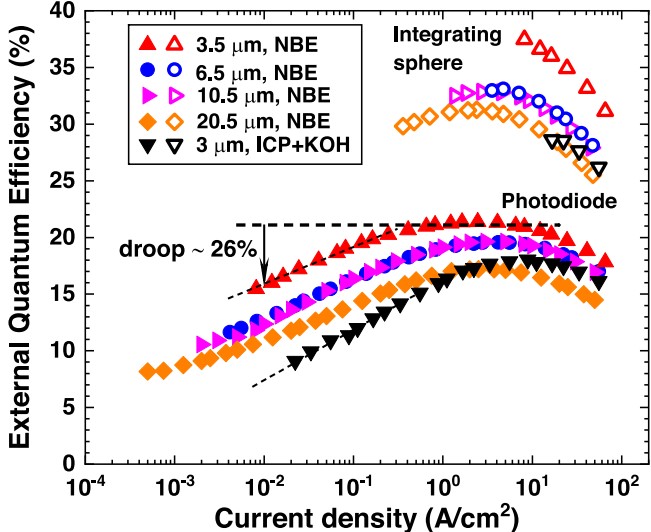

**Fig. 2 | Current density dependence of EQE.** EQE of micro-LEDs fabricated by the NBE process with different sizes ranging from 3.5 × 3.5 μm² to 20.5 × 20.5 μm² as a function of current density is shown. Here, a uniform current density over the whole mesa area was assumed since the spacing between the edge of the metal contact and the edge of the chip (approximately 0.75 μm, see Fig. 1c) is small enough than the current spreading length (see Fig. 3c), and the chip size measured from SEM observation was used to calculate the current density. A 3 × 3 μm² device fabricated by the ICP process and treated with a KOH solution was also given for comparison. The solid up-pointing triangle (open up-pointing triangle), solid circle (open circle), solid right-pointing triangle (open right-pointing triangle), and solid diamond (open diamond) represent the EQEs calculated from the light emission power measured using a photodiode (an integrating sphere) for the 3.5 × 3.5 μm², 6.5 × 6.5 μm², 10.5 × 10.5 μm², and 20.5 × 20.5 μm² micro-LEDs fabricated by the NBE process, respectively. The solid and open down-pointing triangles represent the EQEs calculated from the light emission power measured using a photodiode and integrating sphere, respectively, for the 3 × 3 μm² micro-LED fabricated by the ICP process.

vacuum, $h$ is the Planck constant, and $e$ is the elementary charge. All four EQE curves of the NBE samples exhibited a maximum at a current density of approximately 3 A/cm². The current density at which EQE peaks is indicative of the nonradiative recombination velocity and can be used as a figure of merit to evaluate LED efficiency[20,21]. A lower current density at the peak EQE implies a lower nonradiative recombination rate and thus, a higher EQE for devices with similar radiative and Auger recombination rates. Therefore, the observation of a similar current density at the peak EQE for all the four micro-LEDs fabricated by the NBE process suggests that all the four devices have a similar nonradiative recombination rate. Here, we assumed that all the devices have a similar Auger recombination rate. This is a reasonable assumption because Auger recombination rates of GaN blue LEDs have been shown to be nearly size-independent for LEDs smaller than 100 μm²[22]. Moreover, the peak EQE was found to increase as the chip size decreased. The peak EQE of the 3.5 × 3.5 μm² chip was calculated to be as high as 37.5% from the total emission power measured using the integrating sphere. Although the differences in light-extraction efficiency need to be considered, the aforementioned EQE value is substantially higher than those reported for GaN blue micro-LEDs with comparable dimensions[20]. However, the most notable finding revealed in Fig. 2 is that all NBE-etched devices exhibited very slow decreases in EQEs with decreasing current density from the current density at peak EQEs. We defined an efficiency droop as $(EQE_{peak} - EQE_{0.01A/cm^2})/EQE_{peak}$ to quantitatively evaluate the efficiency decrease in the low current density region, where $EQE_{peak}$ and $EQE_{0.01A/cm^2}$ are the peak EQE and the EQE at a current density of 0.01 A/cm², respectively. The efficiency droop for the 3.5 × 3.5 μm²,

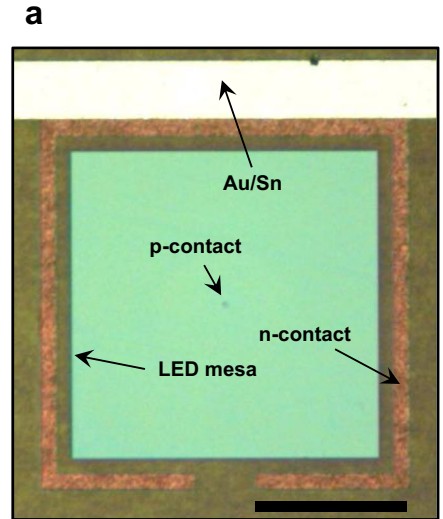

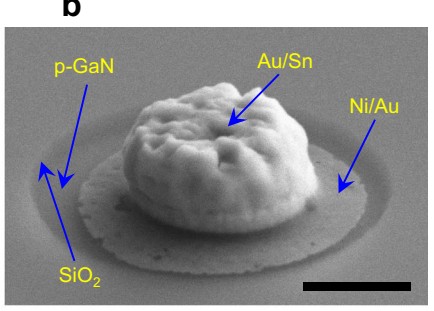

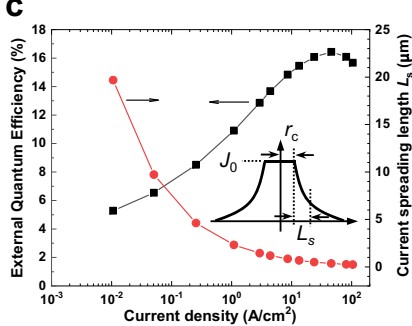

**Fig. 3 | 200 × 200 µm² LED with a small p-contact. a** A microscopic photograph of the 200 × 200 µm² LED with a small Ni/Au p-contact (3 µm in diameter) to investigate the intrinsic EQE of the epitaxial wafer is shown. A stripe-shaped n-contact surrounding the 200 µm mesa was employed. Scale bar, 100 µm. **b** A high-magnification 30° tilted SEM image showing the details of the p-contact. Scale bar, 1 µm. **c** Measured EQE (the black curve) and calculated current spreading length $L_s$ (the red curve) as a function of current density. The inset illustrates schematically the variation of current density along the radial direction, where $J_0$, $r_c$, and $L_s$ represent the current density at the edge of the p-contact, the radius of the p-contact, and the current spreading length, respectively.

$6.5 \times 6.5\,\mu m^2$, $10.5 \times 10.5\,\mu m^2$, and $20.5 \times 20.5\,\mu m^2$ micro-LEDs fabricated by the NBE process were calculated from Fig. 2 to be approximately 26%, 34.7%, 36.7%, and 37.2%, respectively. The findings of this study reveal a surprising result that the efficiency droop decreases with decreasing chip size. The $3.5 \times 3.5\,\mu m^2$ micro-LED exhibited the lowest efficiency droop of approximately 26%. This observation contrasts with the behavior of conventional ICP-etched GaN micro-LEDs, where the efficiency droop typically increases rapidly with chip size reduction below 10 µm because of the enhanced sidewall nonradiative recombination, even with KOH treatment[9,10,23]. These results indicate that sidewall nonradiative recombination, which typically increases when the chip size decreases, was effectively suppressed in the micro-LEDs fabricated using the NBE process to a level that can be neglected relative to the bulk nonradiative recombination of the epitaxial wafer. In other words, the emission efficiency is only limited by the bulk nonradiative recombination which should be size-independent. Considering a size-independent nonradiative recombination rate (a similar IQE) in NBE-etched micro-LEDs, the increase in peak EQEs when the chip size decreases, can be explained by an increase in light-extraction efficiencies in smaller micro-LEDs, which resulted from enhanced sidewall light extraction[20,23]. On the other hand, the $3 \times 3\,\mu m^2$ micro-LED fabricated by the ICP process (Supplementary Fig. 5) exhibited a lower peak EQE of approximately 28.5%, a higher current density at peak EQE of approximately 9 A/cm², and a much larger efficiency droop of approximately 60% at a current density of 0.01 A/cm² compared to the NBE-etched devices. These differences clearly indicate the existence of strong sidewall nonradiative recombination in the ICP-etched device, and that KOH treatment cannot entirely remove sidewall nonradiative recombination, consistent with previous reports[12,20,23–25].

To further confirm the influence of sidewall nonradiative recombination, we examined the intrinsic EQE of the epitaxial wafer by fabricating a 200 × 200 µm² LED with a small (3 µm in diameter) p-contact at the center, using the same processes as those employed for the

micro-LEDs in Fig. 2 (Supplementary Fig. 6). The microscopic photograph of the 200 × 200 µm² LED and the 30° tilted high-magnification SEM image of the 3-µm diameter p-contact are shown in Fig. 3a, b, respectively. In this case, the current is injected into the device through the p-contact and spreads out from the p-contact with a diffusion length $L_s$, which is defined by the following equation:

$$t = \rho L_s \left( r_c + \frac{L_s}{2} \right) \left( J_0 \frac{e}{n_{\text{ideality}} kT} \right) \ln\left( 1 + \frac{L_s}{r_c} \right) \qquad (2)$$

where $t$ represents the thickness of the p-type layer, $\rho$ denotes the resistivity of the p-type layer, $r_c$ denotes the radius of the p-contact metal, $J_0$ represents the current density at the edge of the p-contact, which can be calculated from the injection current $I$ by $I/\pi(r_c + L_s)^2$ (see inset of Fig. 3c), $n_{\text{ideality}}$ is the ideality factor of the LED, $e$ is the elementary charge, $k$ is the Boltzmann constant, and $T$ is the temperature[26]. Additionally, $\rho$ was measured by Hall measurement to be approximately 18 Ωcm. $n_{\text{ideality}}$ was determined to be approximately 2.18 from the I–V curve (Supplementary Figs. 7 and 8). The radius of the Ni/Au p-contact was 1.5 µm ($r_c$). We assumed that sidewall nonradiative recombination was completely avoided if the current spreading length was much shorter than the distance between the p-contact and the mesa edge (approximately 100 µm). Consequently, the measured EQE solely reflected the intrinsic emission efficiency of the epitaxial wafer.

Figure 3c shows the current density dependence of the calculated current spreading length and measured EQE. The EQE reached a maximum at a current density of approximately 40 A/cm², which was one order of magnitude larger than that observed in Fig. 2. This was due to the fact that the current density decreases outside the p-contact with increasing distance from the p-contact edge, as depicted in the inset of Fig. 3c. When the current density under the p-contact is increased to the value at peak EQE in Fig. 2, the local EQE underneath the p-contact will reach a maximum. However, further current

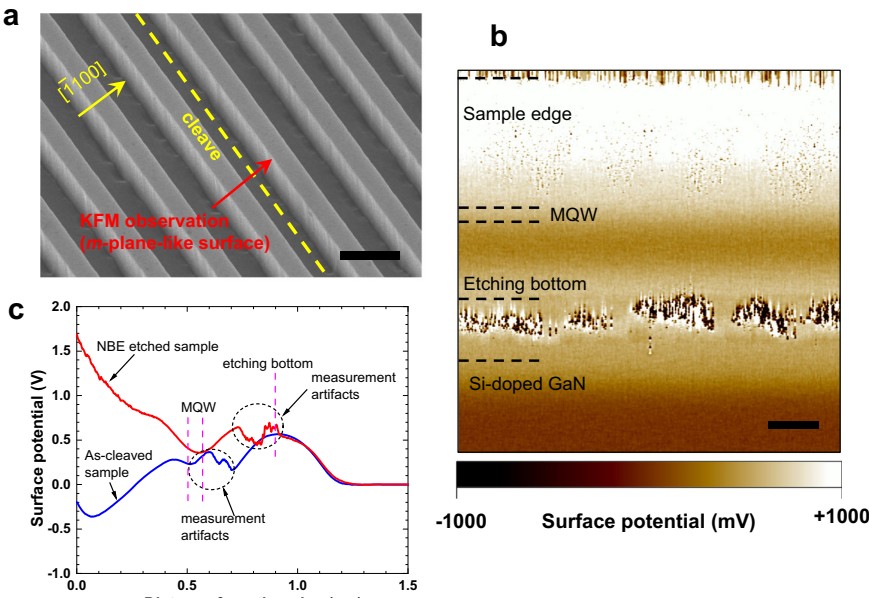

**Fig. 4 | Surface potential measurement by KFM. a** A tilted SEM image of the 1-μm-pitched grating for KFM observation is shown. The dashed line indicates the cleave direction. Scale bar, 1 μm. **b** A typical measured surface potential image is shown. The scan area was 1.5 μm × 1.5 μm. The MQW and Si-doped GaN positions were marked based on growth thickness. The etching bottom position was marked using the NBE etching depth obtained from SEM observation. Scale bar, 200 nm. **c** Surface potential as a function of distance from the edge of the NBE-etched (the red curve) and as-cleaved (the blue curve) reference samples. The small fluctuations marked by dashed circles are due to measurement artifacts.

injection was needed for the position-averaged EQE to reach a maximum, as the EQE outside the p-contact decreased with distance from the p-contact edge. As a rough approximation, we assumed that the EQE vs. current density curve shifted by one order of magnitude towards a higher current density with respect to those given in Fig. 2, where the current density was spatially uniform. Therefore, the efficiency droop at the current density of 0.1 A/cm² in Fig. 3c which was estimated to be approximately 55% are considered corresponding to that at the current density of 0.01 A/cm² in Fig. 2. The current spreading length at the current density of 0.1 A/cm² (approximately 6.3 μm) was much shorter than the distance between the p-contact and the mesa edge (100 μm), so we assumed that the influence of sidewall nonradiative recombination could be neglected, and that the above efficiency droop represented the intrinsic EQE characteristics of the epitaxial wafer.

Considering the uncertainties related to the non-uniform current density distribution in the 200 × 200 μm² device, the above efficiency droop is in reasonable agreement with that of the 20.5 × 20.5 μm² micro-LED in Fig. 2. This suggests that the efficiency droops of the NBE-etched micro-LEDs shown in Fig. 2 were solely determined by non-radiative recombination in the InGaN/GaN bulk layers, and that sidewall-related nonradiative recombination did not give rise to any further reduction in the emission efficiency. This was a surprising conclusion, as nonradiative recombination related to intrinsic sidewall surface states is generally believed to inevitably exist. Nonetheless, our results suggest that both nonradiative recombination induced by sidewall damage generated during mesa etching and that induced by intrinsic sidewall surface states resulting from surface dangling bonds in micro-LEDs fabricated by the NBE process were suppressed to a level that made the influence of sidewall nonradiative recombination on micro-LED efficiencies negligible compared to bulk nonradiative recombination.

## Surface potential measurement of NBE-etched sidewall surfaces

Intrinsic surface states induced by dangling bonds can cause surface band bending, which can be characterized using KFM[27,28]. In this study,

we investigated the surface band bending of the NBE-etched surface by KFM, using the sample shown in Fig. 4a. The wafer was grown on a (0001) *c*-plane n-type freestanding GaN substrate with a resistivity of approximately 0.025 Ω cm, using MOVPE. The substrate was misoriented 0.4° towards the [1-100] direction. The layer structure consisted of 0.5-μm-thick Si-doped GaN layer (electron concentration of approximately $3 \times 10^{18}$ cm⁻³), 0.5-μm-thick unintentionally doped GaN layer (n-type conductivity), a 5-period InGaN/GaN blue-emitting MQW layer (total thickness of approximately 80 nm), and 0.5-μm-thick unintentionally doped GaN layer. The sample was subsequently processed into a 1-μm-pitched grating with equal line and space widths, using photolithography and NBE etching (Fig. 4a). The etching depth is approximately 0.9 μm. The sample was then cleaved along the line-stripe direction, and KFM observations were performed on the exposed *m*-plane-like surface etched by NBE. An as-cleaved sample without a grating pattern was measured as a reference. Figure 4b shows the measured surface potentials of the NBE-etched samples. Distorted images appearing just beneath the etching bottom position are attributed to dimples observed around the corner between the vertical *m*-plane-like surface and the bottom *c*-plane in the SEM image shown in Fig. 4a.

Although a uniform potential image was observed along the grating stripe direction, the surface potential tended to increase from the MQW position toward the sample edge. In Fig. 4c, we present the surface potentials of the NBE-etched sample and the as-cleaved reference sample as a function of the distance from the sample edge. KFM measures the difference in contact potential between the atomic force microscopy (AFM) probe tip and the GaN surface. This difference can be expressed by $(\chi_{tip} - \chi_{GaN}) - SBB - (E_C - E_F)$, where $\chi_{tip}$ is the electron affinity (or Fermi level) of the AFM tip, $\chi_{GaN}$ is the electron affinity of GaN, SBB is the surface band bending of GaN, $E_C$ is the conduction band bottom of GaN, and $E_F$ is the Fermi level of GaN[27]. Since $\chi_{tip}$, $\chi_{GaN}$, $E_C$, and $E_F$ are the same for the NBE-etched and reference samples, the measured potential difference in Fig. 4c reflects only the difference in the surface band bending between the two samples. The *m*-plane GaN surface is typically

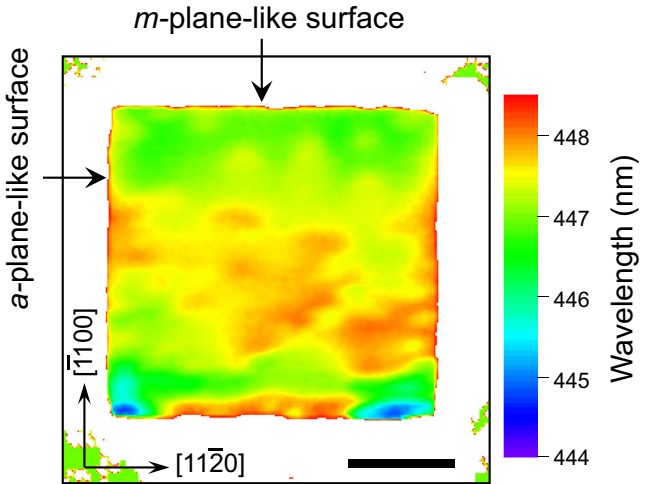

**Fig. 5 | CL mapping of the emission wavelength.** Mapping the CL peak wavelength of a 3.5 × 3.5 μm² micro-LED mesa fabricated by the NBE-process. Scale bar, 1 μm. A purple-to-red color scale was used to represent the wavelength.

reported to show upward band bending (towards the vacuum level) of the order of a few hundred meV[14,28]. In this study, we observed that the NBE-etched sample had a slightly higher (0.3–0.4 V) surface potential and upward surface band bending compared to the reference sample in the region between the etching bottom and MQW, while the two samples exhibited similar surface potentials around the MQW position. This suggests that the NBE-etched sample has a similar or slightly larger number of surface states compared to the reference sample. However, as the measurement position moved towards the sample edge from the MQW position, the surface potential started to increase, and we speculate that this could be due to the adsorption of Cl on the etched surface, because this area was exposed to the Cl neutral beam for a longer time[29]. The KFM results, together with the EQE data presented in Fig. 2, indicate that the surface states on the NBE-etched sidewall surface are not efficient nonradiative recombination centers. Further investigation is necessary to comprehend the unusual behavior of surface states in NBE-etched GaN and InGaN. Nonetheless, we have demonstrated that it is possible to fabricate InGaN/GaN micro-LEDs with negligible sidewall surface nonradiative recombination.

### Mechanism for the low efficiency droop in the 3.5 × 3.5 μm² micro-LED

Finally, we would like to briefly discuss the mechanism behind the observed decrease in efficiency droop with decreasing chip size in micro-LEDs fabricated by the NBE process, as shown in Fig. 2. It is well established that patterning InGaN/GaN MQWs into sub-micrometer-sized nanostructures can lead to strong strain relaxation, which in turn weakens the quantum-confined Stark effect and enhances emission efficiency[30–32]. Strain relaxation is also present in InGaN/GaN mesa structures with diameters of a few to tens of micrometers, with relaxation mainly occurring near the mesa surface over a width of a few hundred nanometers[33,34]. Therefore, we presumably attributed the low-efficiency droop observed in the 3.5 × 3.5 μm² micro-LED to the strain relaxation effect occurring near the mesa surface. To confirm the existence of strain relaxation, we performed cathodoluminescence (CL) mapping of the emission wavelength of the 3.5 × 3.5 μm² mesa fabricated by the NBE process. As shown in Fig. 5, a blue shift of 1–2 nm in the emission wavelength can be clearly observed near the m-plane-like sidewall surface over a width of a few hundred nanometers. This experiment provides direct evidence for the existence of strain relaxation near the mesa sidewall surface of

micro-LEDs studied in this work. However, observation of efficiency increase induced by strain relaxation in the low current density region is challenging in micro-LEDs with strong sidewall nonradiative recombination because the decrease in efficiency caused by sidewall nonradiative recombination can easily offset the efficiency increase induced by strain relaxation. The above discussion indicates again that the sidewall surface nonradiative recombination in micro-LEDs fabricated by the NBE process is negligible compared with the bulk nonradiative recombination.

## Discussion

In summary, we successfully demonstrated a 3.5 × 3.5 μm² InGaN/GaN blue micro-LED with negligible sidewall surface nonradiative recombination. This was achieved through an ultralow-damage NBE etching process. The micro-LED exhibited an EQE droop as low as 26% at a current density of 0.01 A/cm², compared with the peak EQE observed around 3 A/cm². This efficiency droop is approximately 11% lower than that of a 20.5 × 20.5 μm² device. With these results, we anticipate that it will be possible to reduce the chip size even further to meet the requirements of AR smart glasses, because the mesa sidewall does not give rise to any reduction in micro-LED efficiencies.

## Methods

### NBE experiments

For the Cl₂-based etching process, the ICP source power was set to 800 W and modulated at a frequency of 10 kHz with a 50% duty ratio. A Cl₂ gas flow rate of 40 sccm was used as the etching gas, and the pressure in the etching chamber was 0.1 Pa. To enhance the evaporation of the In-containing etching products, the sample stage temperature was set to 130 °C. Additionally, a bias power of 6 W was applied to the carbon aperture in order to control the kinetic energy of the neutral beam. The etching rate achieved under these conditions for the LED wafer used in this study was approximately 5 nm/min, though it is worth noting that the InGaN well layer may exhibit slower etching rates due to the low volatility of In-containing etching products.

### ICP experiments and KOH etching

In the ICP experiments, an ICP machine (RIE-400iPS, Samco Inc.) was utilized. A mixture of Cl₂ and BCl₃ gases was used as the etching gas with flow rates of 50 sccm and 6 sccm, respectively. The ICP and RF bias powers were set to 150 W and 5 W, respectively. The etching pressure and stage temperature were 1 Pa and 20 °C, respectively, resulting in an etching rate of approximately 20 nm/min for the LED wafer used in this study. After ICP etching of the micro-LED mesa, the sample was treated with a 48% KOH solution at approximately 25 °C for 35 min to remove ICP-induced surface damage.

### KFM measurements

KFM measurements were conducted using a Bruker NanoscopeV/Dimension Icon Glovebox AFM in a high-purity argon gas atmosphere at room temperature, where the residual concentrations of both water and oxygen were approximately 0.1 ppm. An Si cantilever covered with Pt/Ir was used as the probe. After cleaving the NBE-etched sample along the grating stripe direction, SEM observation was performed to find a surface area with minimal height differences around the etching bottom. A bias was applied to adjust the surface potential in the starting area (approximately 1.5 μm from the sample edge) to zero, enabling a comparison between the two samples. As the a-plane GaN surface is more difficult to cleave, KFM measurements were performed only on the m-plane.

### CL mapping

A CL mapping experiment was conducted using a Schottky-type field-emission SEM machine (JEOL, JSM-7100F). An acceleration voltage of

8 kV, giving rise to a penetration depth of approximately 450 nm in GaN, was used. The beam current was set at 0.5 nA. An epitaxial wafer with the same layer structure as that used for micro-LED fabrication grown on a planar sapphire substrate was used because the patterned substrate will cause scattering of light and lower the spatial resolution of the mapping. The emission wavelength was measured to be approximately 451 nm by photoluminescence excited by a 375-nm laser (excitation power density: 7.5 W/cm²). An SiO₂ passivation layer was not deposited on the sample surface to avoid the charge-up effect.

## Data availability
The source data underlying all figures presented in the main manuscript and Supplementary Information are provided in the Figshare repository at https://doi.org/10.6084/m9.figshare.23961723.

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

## Acknowledgements

We would like to express our gratitude to Dr. Naoto Kumagai for his valuable discussions on KFM measurements, Dr. Hisashi Yamada for his advice on MOVPE growth on free-standing GaN substrates, and Dr. Reiko Azumi for her continuous support of this work. Dr. Toshikazu Yamada and Ms. Akiko Murai also provided significant help in wire bonding the micro-LED chips.

## Author contributions

S.S. invented the neutral beam technique. X.W. and S.S. jointly proposed applications of the neutral beam technique to micro-LED processing. X.W. performed most of the device fabrication and characterization experiments and prepared the manuscript. X.Z. conducted the majority of the SEM observations, while T.T. performed the MOVPE growth for the KFM measurements. D.O. carried out part of the NBE etching experiments. X.W., S.S. and D.O. participated in extensive discussions on the experimental results.

## Competing interests
The authors declare no competing interests.
