## [Peer Review File · Nature Communications]

REVIEWER COMMENTS

Reviewer #1 (Remarks to the Author):

This is a very interesting manuscript reporting very good achievements related to minimising surface damage from processing and strongly reducing surface recombination. As such, the paper will be of significance to the field of microLEDs.

I have a couple of points of criticism which I would like to see taken care of:

1) In principle, as a scientist, I am not so happy to read (as in the title) "...mesa sidewalls free from non radiative recombination", since I mean that there is no such case. Depending on how closely you look, you will always observe such effects. Rather I would like to replace this kind of vague statements by a quantification such as an upper limit of non radiative recombination, or a statement of how drastically the improvement is in terms of reduction of the non radiative recombination.

2) I think it is unsatisfactory to notice that the authors never measured the efficiency of the devices using a standard integrating sphere, rather just from measuring the emission with a calibrated silicon photodiode and then estimate the EQE based on estimation of the light collection angle and the assumption of a Lambertian emission pattern (an assumption they do not support by measurements). Also, it is known that the emission patterns can vary dramatically with different chip sizes.

Reviewer #2 (Remarks to the Author):

The manuscript describes the use of NBE to result in damage-free microLEDs. The objective of the manuscript is interesting to the semiconductor community. However, the data presented in the manuscript requires more work to be considered for acceptance. Below are the comments.

1. The authors claimed that the four vertical smooth sidewall surfaces correspond to the non-polar planes of GaN. However, it is unclear that how the authors achieve this conclusion. Vertical and smooth sidewall surfaces can be achieved by varying the etch parameters, such as power and chemistry, yielding smooth sidewall surfaces does not mean the exposure of non-polar planes. This paper (Kuritzky et al 2016 Semicond. Sci. Technol. 31 075008) showed smooth sidewall surfaces can be obtained in semipolar and non-polar GaN substrates. Even with precise alignment, the alignment offset in conventional lithography accuracy does not guarantee the alignment of non-polar plane. This is important to the overall discussion, since surface recombination is dependent on crystal plane.

2. The authors showed that smaller microLEDs have lower efficiency drop before the peak EQE, which they also pointed out this is the opposite trend from the literature. They claimed the nonradiative recombination is suppressed using the NBE process. However, larger LEDs should have less impacts from sidewall defects and its performance should be less dominated by the fabrication techniques. Moreover, the peak current density of all NBE devices showed almost the same value about 7 A/cm², where larger LEDs should have low peak current density. The peak EQE of the larger devices are also lower than the 3.5 μm device, which is confusing. There are several contradictions with the findings in the literature (Olivier et al 2017 J. Lumin. 191 112-116 and Tian et al 2012 APL 231110). The authors have to address this.

3. The study of the 200x200 μm² LED is not very convincing to the overall results. First, the light extraction is different between the two cases, which will affect the EQE measurement. Secondly, the current spreading aspect between microLED and LED is also distinctive, such that the thermal effect might play a bigger role in the 200x200 μm² LED but not in the 3.5x3.5 μm² microLED.

4. The authors claimed strain relaxation plays a role in the low efficiency droop. If so, the authors should provide additional strain relaxation data, such as XRD and/or emission spectra, to demonstrate this. For now, the strain argument is a handwaving explanation. Also, the sizes used in ref 29 are in the nano-scale that is less than 1 μm, which is significantly smaller than the presented device size. The authors should explain how strain is related to the device dimension.

Reviewer #3 (Remarks to the Author):

The authors of this manuscript have demonstrated the fabrication of highly efficient top-down 3.5 μm-sized InGaN LEDs using a damage-free etching process called neutral beam etching. The authors assert that this process effectively suppresses surface defects in semiconductor fabrication. Compared to conventional ICP-etched devices, the proposed Micro-LEDs exhibit a low current density of peak EQE and a decrease in EQE at a current density of 0.01 A/cm², which is only 25%, an exceptionally low value. These micro-LEDs have the potential to significantly enhance energy efficiency, making them crucial for VR/AR microdisplays. However, it is important to note that the results presented in this manuscript are inconsistent with other findings, and the authors have not provided sufficient supporting evidence for their conclusions. The reviewer has raised concerns specifically regarding the accuracy of the experimental EQE measurements. Consequently, based on these concerns, I do not believe this manuscript meets the criteria for publication in Nature Communications.

1. Line 50 of “approximately 50% of the injected ...”: Ref 5 does not provide support for the statement made in Line 50. Therefore, the authors should provide the appropriate reference. Below are the suggested references that could potentially address the intended information.

a. US Department of energy, “Thermal Management of White LEDs (Thermal Management of White LEDs (energy.gov))” (2007),

b. Y Qin, et al., “A Simple Method for Comparative Study on the Thermal Performance of LEDs and Fluorescent Lamps”, IEEE Trans. Power Electron., vol. 25, no. 2, pp. 507–513, Feb. 2010, doi: 10.1109/TPEL.2009.2026752

2. Line 168 of “comparable to those reported for state-of-the-art InGaN/GaN blue LEDs”: Reference 11 does not pertain to the statement made in Line 168. Therefore, the authors should provide the correct references that are relevant to the intended information.

3. Line 171 – 173 of “... a peak EQE as high as 20.8%. Taking into account the collection efficiency of 38% in our measurement system, the actual EQE should exceed 50%”: The reported peak EQE of 20.8% (possibly exceeding 50% in actuality) appears excessively high. Typically, Micro-LEDs exhibit light extraction efficiencies (LEE) ranging from 10% to 20%. Consequently, devices with EQEs surpassing 20% should have internal quantum efficiencies (IQEs) exceeding 90%. However, achieving an IQE above 95% is not realistic. To illustrate, a device with an EQE of 50% and an IQE of 95% would necessitate an LEE of approximately 53%, which is unattainable for non-packed LED devices. Even for an EQE of 20.8% and an IQE of 95%, an LEE of 22% would be required, which is challenging to achieve. Therefore, accurately estimating a peak EQE of 20.8% is of critical importance in this study. It is highly recommended to compare the EQEs of the devices in the manuscript with EQE measurements conducted using an integrating sphere, as this approach can provide accurate EQE values for devices larger than 10 μm . This calibration process can ensure accurate EQE measurements for the devices discussed in the manuscript.

4. Line 214 – 215 of “ideality factor was determined to be approximately 2.18 from I-V curves”: The observed ideality factor of 2.18 appears unusually large for LEDs with exceptionally good EQEs. According to the Shockley model, an ideality factor of 2 is typically associated with Shockley-Read-Hall (SRH) recombination through traps. If the ideality factor exceeds 2, it could be indicative of deep-level traps or a dominant tunneling mechanism involving non-radiative recombination centers (refer to JM Shar et al., J Appl. Phys, 94, 2627 (2009), and D. Zhu et al., APL, 94, 081113 (2009)). Therefore, the ideality factor of 2.18 suggests the presence of significant deep traps affecting the device performance, which sharply contrasts with the notable suppression of non-radiative recombination by traps in the NBE-processed devices. Considering that the NBE process facilitates damage-free etching on the surface of the active layers, it raises questions regarding the physical origin of the high ideality factor. Additionally, it is essential for the authors to present the I-V curves for all devices. It is highly recommended to obtain ideality factors for all devices and analyze them in connection with the EQEs.

5. Line 226 of “as depicted in the inset of Fig. 3c”: there is no inset in Fig. 3c.

6. Line 253 – 298 of Surface potential measurements: The authors employed KFM to measure the upward bending of the surface band. However, according to Ref. 15 by F. Jiang et al., the presence of intrinsic surface states is indicated by this upward bending. Ref. 15 explicitly states, "As the Micro-LED size decreases, the hole accumulation at the surface becomes more serious. The accumulated holes resulting from the upward bending of the surface energy band significantly impact the IQE and peak current density." Consequently, the authors of Ref. 15 suggested minimizing the upward bending of the surface band as a means to prevent hole accumulation and mitigate the influence of intrinsic surface states. This approach would lead to the suppression of these states.

7. Line 300 – 315 of “Mechanism for the decrease in efficiency droop with decreasing chip size in NBE-etched micro-LEDs”: In Figure 2, it is observed that the EQEs of smaller devices exhibit higher values. This phenomenon can potentially be explained by improved current spreading in smaller devices (for example, see the reference H. Yu et al., Opt Lett 3271 (2021)). Consequently, the lower efficiency droop observed in smaller devices might also be attributed to enhanced current spreading due to their smaller size, although this connection remains unclear at present. To investigate the suggested strain relaxation near the mesa surface proposed by the authors, micro-Raman spectroscopy measurements can be conducted (for example, see the work by T. Rupp et al., Thin Solid films 294, 27 (1997)), to identify strain relaxation effects. Implementing micro-Raman spectroscopy measurements would provide valuable insights into the proposed strain-relaxation mechanism near the mesa surface.

8. To establish definitive evidence of defect-free characteristics resulting from the NBE process, it is essential to conduct systematic studies using deep level transient spectroscopy (DLTS). However, it should be noted that performing DLTS on small-sized devices can be a challenging task.

Response letter to the first reviewer

Dear reviewer

Thank you for your careful review of our manuscript and insightful comments. We revised the manuscript according to your comments. Please find a summary of the changes we made in the revised manuscript. The changes are highlighted using a blue font in the revised manuscript.

Comment #1

In principle, as a scientist, I am not so happy to read (as in the title) "...mesa sidewalls free from non radiative recombination", since I mean that there is no such case. Depending on how closely you look, you will always observe such effects. Rather I would like to replace this kind of vague statements by a quantification such as an upper limit of non radiative recombination, or a statement of how drastically the improvement is in terms of reduction of the non radiative recombination.

Response

Indeed, we agree that the data presented in this manuscript only suggest that sidewall nonradiative recombination was suppressed to a level comparable to or lower than bulk nonradiative recombination. In other words, the influence of sidewall nonradiative recombination on micro-LED efficiency can be neglected compared with bulk nonradiative recombination. The data do not assure the absence of sidewall nonradiative recombination. We changed the expression "with mesa sidewalls free from nonradiative recombination" to "with negligible sidewall surface nonradiative recombination" or "with sidewall nonradiative recombination that can be neglected compared with bulk nonradiative recombination," both in the title and in the main text.

Please see, the title, page 2 (the abstract), page 5 (second paragraph), page 12 (first paragraph), and page 15 (the first paragraph and the conclusions).

Comment #2

I think it is unsatisfactory to notice that the authors never measured the efficiency of the devices using a standard integrating sphere, rather just from measuring the emission with a calibrated silicon photodiode and then estimate the EQE based on estimation of the light collection angle and the assumption of a Lambertian emission pattern (an assumption they do not support by measurements). Also, it is known that the emission patterns can vary

dramatically with different chip sizes.

Response

We did not measure the light output power by using an integrating sphere in the original manuscript because the light output power, especially that in the low current density region that we focused on in this study, was too weak (1 nW - a few μ W for the 3.5- μ m micro-LED) to be measured using a standard integrating sphere. In the revised manuscript, we included measurements of the light output power under high current densities acquired using a small (2 inch) integrating sphere (Thorlabs, S142C) with a detection limit of 1 μ W. Figure 1 displays a photograph of the integrating sphere, in which an LED package was fixed to the input aperture.

In Fig. 2 of the revised manuscript, we present both the EQEs calculated from light output power measured using a photodiode and integrating sphere. The EQEs measured using the integrating sphere was 1.7-1.8 times larger than those measured by the photodiode for micro-LEDs fabricated via the NBE process. EQEs calculated from the photodiode measurement were used to discuss the efficiency reduction characteristics in the low current density region, while those calculated from the integrating sphere measurements were used to estimate the peak EQE.

In the revised manuscript, we added information of integrating sphere measurements under the experimental section (page 7, the second paragraph) and discussion of the peak EQE of the 3.5- μ m micro-LED based on integrating sphere measurement (page 8, the central part).

Fig.1 Photograph of the 2-inch integrating sphere with an LED package fixed to the input aperture.

Other major changes

- 1) The 3.5- μm NBE and the 3- μm ICP devices used in the original manuscript were broken during the integrating sphere measurement. The data measured from a new chip were used in the revised manuscript.
- 2) We performed wavelength mapping of a 3.5- μm micro-LED mesa by cathode luminescence to confirm the existence of strain relaxation in a 3.5- μm micro-LED, as suggested by other reviewers. The result is given in Fig. 5 of the revised manuscript.

We hope the above revisions answer your questions and address the concerns raised in the comments. We appreciate your consideration to publish our manuscript.

Response letter to the second reviewer

Dear reviewer

Thank you for carefully reviewing our manuscript and useful comments. We revised the manuscript according to your comments. Please find a summary of the changes we made in the revised manuscript. The changes we made were highlighted by blue letters in the revised manuscript.

Comment #1

The authors claimed that the four vertical smooth sidewall surfaces correspond to the non-polar planes of GaN. However, it is unclear that how the authors achieve this conclusion. Vertical and smooth sidewall surfaces can be achieved by varying the etch parameters, such as power and chemistry, yielding smooth sidewall surfaces does not mean the exposure of non-polar planes. This paper (Kuritzky et al 2016 Semicond. Sci. Technol. 31 075008) showed smooth sidewall surfaces can be obtained in semipolar and non-polar GaN substrates. Even with precise alignment, the alignment offset in conventional lithography accuracy does not guarantee the alignment of non-polar plane. This is important to the overall discussion, since surface recombination is dependent on crystal plane.

Response

We assigned the sidewall surfaces to the m - and a -planes of the GaN crystal based on the crystalline orientation of the epitaxial wafer. We apologize for failing to include the information in the original manuscript. We included the relevant information in the revised manuscript (page 6, the bottom 5 lines). We also added the crystalline direction in Fig. 1(c). We believe that these planes are close to the m - and a -planes of GaN, but we agree with you that there should be some deviation from the exact m - and a -planes of GaN owing to limited precision of the device processes. We also added one sentence to mention the existence of possible misalignment (page 6, the bottom 2 lines).

Comments #2

The authors showed that smaller microLEDs have lower efficiency drop before the peak EQE, which they also pointed out this is the opposite trend from the literature. They claimed the nonradiative recombination is suppressed using the NBE process. However, larger LEDs should have less impacts from sidewall defects and its performance should be less dominated by the fabrication techniques. Moreover, the peak current density of all

NBE devices showed almost the same value about 7 A/cm², where larger LEDs should have low peak current density. The peak EQE of the larger devices are also lower than the 3.5μm device, which is confusing. There are several contradictions with the findings in the literature (Olivier et al 2017 J. Lumin. 191 112-116 and Tian et al 2012 APL 231110). The authors have to address this.

Response

The main conclusion of the present paper is that sidewall nonradiative recombination was suppressed to a negligible level compared with bulk nonradiative recombination via the neutral beam technique. In other words, nonradiative recombination is only limited by bulk nonradiative recombination (or the intrinsic nonradiative recombination), which should be size independent. Accordingly, it is reasonable to observe a size-independent current density at the peak EQE. The increase in peak EQE with decreasing the chip size can be explained by an increase in the light-extraction efficiency with a decrease in the chip size, as discussed in Refs. 19, 21, primarily because of the enhanced sidewall light scattering.

In the revised manuscript, we modified the text to describe the above discussion more clearly. Additionally, we also included the influence of light-extraction efficiency. Please see, page 9, the central part.

The lower efficiency droop rate in the low current density region below the peak EQE observed in the 3.5-μm micro-LED is explained by the strain relaxation effect, which will be discussed in the response to your last comment.

Comment #3

The study of the 200x200 μm² LED is not very convincing to the overall results. First, the light extraction is different between the two cases, which will affect the EQE measurement. Secondly, the current spreading aspect between microLED and LED is also distinctive, such that the thermal effect might play a bigger role in the 200x200 μm² LED but not in the 3.5x3.5 μm² microLED.

Response

Indeed, the 200 × 200 μm² LED experiment contains some uncertainties because some approximations were used. However, we believe it is a useful result in the sense that it provides a guideline to the intrinsic EQE vs current density characteristics of the epitaxial

wafer used in this study. We would like to address your concerns as follows:

Difference in light-extraction efficiency:

The light-extraction efficiency of the 200- μm device is indeed lower than those of micro-LEDs, which can be clearly confirmed from the peak EQE values (Fig. 2, Fig. 3c). However, because we only concerned about the decrease of EQE with the decreasing current density, but not the absolute value of EQE, the difference in light-extraction efficiency does not influence the discussion of the present paper.

Thermal effect

Since a similar sized p-contact was used in the 200- μm LED (2.5 μm in diameter circle) as compared with the 3.5- μm micro-LED (2 \times 2 μm^2 square), the electrical power injected to the two devices (or the amount of heat generation) under a certain current density is almost the same for the two devices. Moreover, in the 200- μm LED, apart from efficient heat dissipation through the AuSn eutectic metal to the Si submount, heat can also dissipate laterally through the GaN layer, which has a high thermal conductivity (200 mW/cm \cdot K). Therefore, we believe that thermal effect will not give rise to a meaningful influence on the measurement of the EQE values.

Comment #4

The authors claimed strain relaxation plays a role in the low efficiency droop. If so, the authors should provide additional strain relaxation data, such as XRD and/or emission spectra, to demonstrate this. For now, the strain argument is a handwaving explanation. Also, the sizes used in ref 29 are in the nano-scale that is less than 1 μm , which is significantly smaller than the presented device size. The authors should explain how strain is related to the device dimension.

Response

Thank you for the question. To confirm the existence of strain relaxation near the mesa edge, we measured the cathode luminescence (CL) mapping of the emission wavelength of a 3.5- μm micro-LED mesa fabricated via the NBE process. The result was added to the manuscript as Fig. 5. Indeed, a blue shift of approximately 2 nm can be observed near the edge of the *m*-plane sidewall surface over a width of a few hundred nm. We attribute this result as evidence of the existence of strain relaxation in the 3.5- μm micro-LED. Please refer to page 14, the bottom part, line 326-333.

As you pointed out, strong strain relaxation was typically observed in nanostructures smaller than 1 μm . However, strain relaxation near the edge of large mesas (a few to a few tens of μm) has also been reported by several groups (Ref. 32, 33(a new reference)). Our observation agrees with these reports.

Other major changes

- 1) “with mesa sidewalls free from nonradiative recombination” was changed to “with negligible sidewall surface nonradiative recombination” in the title and in the main text, as suggested by another reviewer.
- 2) The light output power of micro-LEDs under high current densities was measured by using a 2-inch integrating sphere to obtain a more precise estimation of the absolute EQE. The results have been added to Fig. 2.
- 3) The 3.5- μm NBE and the 3- μm ICP devices used in the original manuscript were broken during the integrating sphere measurement. The data measured from a new chip were used in the revised manuscript.

We hope the above revisions could answer your comments and we appreciate your consideration on the publication of our manuscript.

Response letter to the third reviewer

Dear reviewer

We appreciate your careful review of our manuscript and valuable comments to improve our manuscript. We revised the manuscript according to your comments. Please find following a summary of the changes we made in the revised manuscript. The changes we made have been highlighted by blue letters in the revised manuscript.

Comment #1

Line 50 of “approximately 50% of the injected ...”: Ref 5 does not provide support for the statement made in Line 50. Therefore, the authors should provide the appropriate reference. Below are the suggested references that could potentially address the intended information.

- a. US Department of energy, “Thermal Management of White LEDs (Thermal Management of White LEDs (energy.gov))” (2007),
- b. Y Qin, et al., “A Simple Method for Comparative Study on the Thermal Performance of LEDs and Fluorescent Lamps”, IEEE Trans. Power Electron., vol. 25, no. 2, pp. 507–513, Feb. 2010, doi: 10.1109/TPEL.2009.2026752

Response

Thank you for your suggestion of new references. However, we found that these papers deal with phosphor-converted white LEDs whose efficiencies (electric power to light conversion efficiency) are lower than those of monochromatic LEDs owing to the energy loss induced by phosphor conversion. The power conversion efficiency referred in “paper a” was 15% – 25%. On the other hand, for display applications, we should be concerned about the efficiency of monochromatic red, green, and blue LEDs, but not white LEDs. According to Fig. 1 of Ref. 5, the external quantum efficiencies of blue (460 nm), green (525 nm), and red (630 nm) LEDs were estimated to be 70%, 40%, and 64%, respectively. Therefore, we believe that 50% is a reasonable representation of the quantum efficiencies of monochromatic RGB LEDs.

Meanwhile, it is true that thermal management is crucial to suppress the temperature increase of an LED package induced by heat generation. Unfortunately, it would be difficult to implement an active thermal dissipation design in AR smart glasses owing to the strict light-weight requirement, which makes low current density operation of micro-LEDs crucial for AR microdisplays.

Based on the above discussion, we revised the manuscript as follows.

We indicated that the “50% efficiency” is the electric power to light conversion efficiency of monochromatic red, green, and blue LEDs. Moreover, we added sentences to mention the importance of thermal management and explain why it is difficult to design thermal dissipation in AR displays. We added the first reference you suggested as a new reference (Ref. 6 of the revised manuscript). Please see, page 3, the first paragraph.

Comment #2

Line 168 of “comparable to those reported for state-of-the-art InGaN/GaN blue LEDs”: Reference 11 does not pertain to the statement made in Line 168. Therefore, the authors should provide the correct references that are relevant to the intended information.

Response

We apologize for the mistake and appreciate the diligence of the reviewer. The correct reference should be Ref. 12 of the original manuscript (Ref. 13 of the revised version).

However, in the revised manuscript, we deleted the comparison of current densities at the peak EQE of micro-LEDs fabricated in this work with those in the above reference.

Comment #3

Line 171 – 173 of “... a peak EQE as high as 20.8%. Taking into account the collection efficiency of 38% in our measurement system, the actual EQE should exceed 50%”: The reported peak EQE of 20.8% (possibly exceeding 50% in actuality) appears excessively high. Typically, Micro-LEDs exhibit light extraction efficiencies (LEE) ranging from 10% to 20%. Consequently, devices with EQEs surpassing 20% should have internal quantum efficiencies (IQEs) exceeding 90%. However, achieving an IQE above 95% is not realistic. To illustrate, a device with an EQE of 50% and an IQE of 95% would necessitate an LEE of approximately 53%, which is unattainable for non-packed LED devices. Even for an EQE of 20.8% and an IQE of 95%, an LEE of 22% would be required, which is challenging to achieve. Therefore, accurately estimating a peak EQE of 20.8% is of critical importance in this study. It is highly recommended to compare the EQEs of the devices in the manuscript with EQE measurements conducted using an integrating sphere, as this approach can provide accurate EQE values for devices larger than 10 μm . This calibration process can ensure accurate EQE measurements for the devices discussed

in the manuscript.

Response

Measurement of peak EQE using an integrating sphere

We did not measure the light output power using an integrating sphere in the original manuscript because the light output power, especially that in the low current density region that we focused on in this study, was too weak (1 nW - a few μ W for the 3.5- μ m micro-LED) to be measured by an integrating sphere. In the revised manuscript, we measured the light output power under high current densities using a small (2 inch) integrating sphere (Thorlabs, S142C) with a detection limit of 1 μ W. Figure 1 displays a photograph of the integrating sphere, where an LED package is fixed to the input aperture.

We presented in Fig. 2 of the revised manuscript both the EQEs calculated from light output power measured by photodiode and integrating sphere. The EQEs measured by integrating sphere was 1.7-1.8 times larger than those measured by the photodiode for micro-LEDs fabricated via the NBE process. EQEs calculated from the photodiode measurement were used to discuss the efficiency decrease characteristics in the low current density region, while those calculated from the integrating sphere measurement were used to estimate the peak EQE. A peak EQE of 37.5% was obtained for the 3.5- μ m micro-LED. Please see the central part of page 8 of the revised manuscript.

Fig.1 Photograph of the 2-inch integrating sphere with an LED package fixed to the input aperture.

Light-extraction efficiency

The LED structure used in this study was grown on a patterned sapphire substrate which could significantly enhance the light-extraction efficiency of an LED. It has also been reported that reducing the micro-LED size could further improve the light-extraction efficiency, resulting from enhanced scattering of light by sidewall surfaces (Ref. 20 of the revised manuscript). For example, using an LED wafer grown on patterned sapphire substrates, D. Hwang *et al.* reported an EQE of approximately 40% for a $10 \times 10 \mu\text{m}^2$ micro-LED after resin encapsulation¹. Lee *et al.* reported an EQE of 30% for a $5 \times 5 \mu\text{m}^2$ micro-LED (Ref. 23 of the revised manuscript). In the 3.5- μm micro-LED fabricated in this work, assuming a peak IQE of 95%, the measured 37.5% peak EQE corresponds to a light-extraction efficiency of 40%. We believe this efficiency is reasonable if we consider light-extraction efficiency enhancement resulted from patterned substrates and the small chip size.

In the revised manuscript, we added one sentence to explain the influence of chip size on light-extraction efficiency. Please see, page 9, line 201-205.

- 1) D. Hwang et al., “Sustained high external quantum efficiency in ultrasmall blue III-nitride micro-LEDs”, *Appl. Phys. Express* **10**, 032101 (2017).

Comment #4

Line 214 – 215 of “ideality factor was determined to be approximately 2.18 from I-V curves”: The observed ideality factor of 2.18 appears unusually large for LEDs with exceptionally good EQEs. According to the Shockley model, an ideality factor of 2 is typically associated with Shockley-Read-Hall (SRH) recombination through traps. If the ideality factor exceeds 2, it could be indicative of deep-level traps or a dominant tunneling mechanism involving non-radiative recombination centers (refer to JM Shar et al., *J Appl. Phys.*, 94, 2627 (2009), and D. Zhu et al., *APL*, 94, 081113 (2009)). Therefore, the ideality factor of 2.18 suggests the presence of significant deep traps affecting the device performance, which sharply contrasts with the notable suppression of non-radiative recombination by traps in the NBE-processed devices. Considering that the NBE process facilitates damage-free etching on the surface of the active layers, it raises questions regarding the physical origin of the high ideality factor. Additionally, it is essential for the authors to present the I-V curves for all devices. It is highly recommended to obtain ideality factors for all devices and analyze them in connection with the EQEs.

Response

First, the ideality factor measured in this work is typical of that of high-quality InGaN/GaN LEDs. The ideality factor of InGaN/GaN blue LEDs near the turn on voltage region is typically in the range of 1.5 – 2. As an example, in Fig. 2, we presented the ideality factor of a high-quality InGaN/GaN LED grown on a free-standing GaN substrate as a function of current density. You can also find the ideality factors of InGaN/GaN micro-LEDs from Ref. 22 of the revised manuscript, which are in the range of 1.8 – 2.4. As you pointed out, this is extremely different from that of other material systems, such as GaAs and AlGaInP, where the ideality factor should be close to unity. This is probably because that the crystal of GaN is not as perfect as GaAs, AlGaInP, etc.

In Fig. 8 of the Supplementary Information, we included I-V curves of all the micro-LEDs studied in this work. It can be seen that the ideality factors near the turn on voltage region were in the range of 1.5 - 1.9, typical of high-quality InGaN/GaN LEDs. The 200- μm device showed an ideality factor slightly larger than that of micro-LEDs, which can be explained by the factor that the current density of the 200- μm LED decreases faster than the micro-LEDs with a decreasing injection current because the current spreading length increases as the injection current decreases (current density = $I/\pi(r_c+L_s)^2$).

Fig. 2 Ideality factor of an InGaN/GaN blue LED grown on a free-standing GaN substrate (Reproduced from: A. David et al., “Electrical properties of III-nitride LEDs: Recombination-based injection model and theoretical limits to electrical efficiency and electroluminescence cooling”, Appl. Phys. Lett. **109**, 083501 (2016))

Comment #5

Line 226 of “as depicted in the inset of Fig. 3c”: there is no inset in Fig. 3c.

Response

We apologize for failing to include the inset of Fig. 3c in the original version. We added the inset to Fig. 3c in the revised manuscript.

Comment #6

Line 253 – 298 of Surface potential measurements: The authors employed KFM to measure the upward bending of the surface band. However, according to Ref. 15 by F. Jiang et al., the presence of intrinsic surface states is indicated by this upward bending. Ref. 15 explicitly states, "As the Micro-LED size decreases, the hole accumulation at the surface becomes more serious. The accumulated holes resulting from the upward bending of the surface energy band significantly impact the IQE and peak current density." Consequently, the authors of Ref. 15 suggested minimizing the upward bending of the surface band as a means to prevent hole accumulation and mitigate the influence of intrinsic surface states. This approach would lead to the suppression of these states.

Response

The sample used for KFM measurement was not a p - n junction, it was just a non-intentionally doped MQW (n -type conductivity) because the inclusion of a p -type layer may complicate the KFM measurements. Therefore, the increase in surface potential near the top layer cannot be explained by the accumulation of holes from a p -type layer. We cannot precisely explain the reason for the increase in surface potential near the top layer at the moment. Further investigation on surface states of sidewall surface etched by the NBE technique will be required to understand this phenomenon.

Comment #7

Line 300 – 315 of “Mechanism for the decrease in efficiency droop with decreasing chip size in NBE-etched micro-LEDs”: In Figure 2, it is observed that the EQEs of smaller devices exhibit higher values. This phenomenon can potentially be explained by improved current spreading in smaller devices (for example, see the reference H. Yu et al., Opt Lett 3271 (2021)). Consequently, the lower efficiency droop observed in smaller devices might also be attributed to enhanced current spreading due to their smaller size, although this connection remains unclear at present. To investigate the suggested strain relaxation near the mesa surface proposed by the authors, micro-Raman spectroscopy

measurements can be conducted (for example, see the work by T. Rupp et al., Thin Solid Films 294, 27 (1997)), to identify strain relaxation effects. Implementing micro-Raman spectroscopy measurements would provide valuable insights into the proposed strain-relaxation mechanism near the mesa surface.

Response

Current crowding effect

The current crowding effect in *p*-side up InGaN/GaN LEDs grown on sapphire substrate is only important for large-size chips ($>200 \times 200 \mu\text{m}^2$) under high current densities. Please see, for example, the conclusion of X. Guo, E. F. Schubert, Appl. Phys. Lett. **90**, 4191 (2001). In particular, the micro-LED mesas studied in this work were sandwiched between two *n*-contact stripes (Supplementary Information, Fig. 4). In this case, current spreading will occur on both sides of the micro-LED mesa. Therefore, we believe that the current crowding effect can be neglected in the micro-LEDs studied in the present work.

Strain relaxation effect

We attributed the low efficiency droop observed in the 3.5- μm micro-LED to the strain relaxation effect occurring near the mesa edge area. To confirm the existence of strain relaxation near the mesa edge, we performed cathode luminescence (CL) mapping of the emission wavelength of a 3.5- μm micro-LED mesa fabricated via the NBE process. The result has been added to the revised manuscript as Fig. 5. We could observe a blue shift of approximately 2 nm near the edge of the *m*-plane sidewall surface over a width of a few hundred nano meters. We attributed this result as evidence for the existence of strain relaxation in the 3.5- μm micro-LED. Please refer to page 14, the bottom part.

Comment #8

To establish definitive evidence of defect-free characteristics resulting from the NBE process, it is essential to conduct systematic studies using deep level transient spectroscopy (DLTS). However, it should be noted that performing DLTS on small-sized devices can be a challenging task.

Response

Indeed, we are attempting to investigate the defect states of GaN surface etched by the NBE process as a new project. As you pointed out, it is an extremely challenging task. We hope to achieve useful results in the near future.

Other major changes

- 1) “with mesa sidewalls free from nonradiative recombination” was changed to “with negligible sidewall surface nonradiative recombination” in the title and in the main text, as suggested by another reviewer.
- 2) The 3.5- μm NBE and the 3- μm ICP devices used in the original manuscript were broken during the integrating sphere measurement. The data measured from a new chip were used in the revised manuscript.

We hope the above revisions and responses address the points raised in your comments and we appreciate your consideration on the publication of our manuscript.

REVIEWER COMMENTS

Reviewer #1 (Remarks to the Author):

I think that the modified version of the manuscript "3.5 × 3.5 μm² GaN blue micro-LEDs with negligible sidewall surface nonradiative recombination" is now in a good shape and suitable for publishing.

Reviewer #2 (Remarks to the Author):

The response to vertical sidewall does not give good conclusion. Since the authors agree that vertical sidewalls do not represent a/m planes, the conclusion provided by the authors is ambiguous and quite tempting to cause the readers to believe that a/m planes are exposed. Therefore, this part of the results should be revised.

Secondly, the discussion in EQE does not give a satisfied reasoning. Yes, it is understood that the authors wanted to use size-independent EQE to show the nonradiative recombination, or Shockley-Read-Hall (SRH nonradiative recombination) is the same for all sizes. However, the results presented in the manuscript does not fit with the expectation from the typical ABC model. Auger recombination should be higher in larger LEDs, so they should have greater C coefficient compared to the smaller counterparts, which results in a peak EQE at lower peak current density. The fact that all the LEDs have the same maximum EQE at almost current density value, this is somewhat strange from the ABC model. This indicates there is something going on from the internal quantum efficiency aspect, but the authors did not address this.

Reviewer #3 (Remarks to the Author):

In the revised manuscript, the authors have effectively addressed all the concerns raised by both the other two reviewers and myself. As a result, I'm confident that it's almost ready for publication.

Response letter to the second reviewer

Dear reviewer

Thank you for carefully reviewing the revised manuscript and further comments. We revised the manuscript further according to your comments. Please find a summary of the changes we made in the revised manuscript. The changes we made were highlighted by blue letters in the revised manuscript.

Comment #1

The response to vertical sidewall does not give good conclusion. Since the authors agree that vertical sidewalls do not represent a/m planes, the conclusion provided by the authors is ambiguous and quite tempting to cause the readers to believe that a/m planes are exposed. Therefore, this part of the results should be revised.

Response

We agree that ambiguities remained in the first version of the revised manuscript concerning the assignment of the sidewall surfaces. To make this point more explicitly, we changed “*m*-plane”, “*a*-plane”, “nonpolar sidewall surfaces” to “*m*-plane-like surface”, “*a*-plane-like surface”, “nonpolar-like sidewall surfaces” in the present version of the manuscript, both in the main text and figures. Please see, line 141 – line 146, line 286, line 290, line 334, Fig. 1c, Fig. 4a., Fig. 5.

Comments #2

Secondly, the discussion in EQE does not give a satisfied reasoning. Yes, it is understood that the authors wanted to use size-independent EQE to show the nonradiative recombination, or Shockley-Read-Hall (SRH nonradiative recombination) is the same for all sizes. However, the results presented in the manuscript does not fit with the expectation from the typical ABC model. Auger recombination should be higher in larger LEDs, so they should have greater C coefficient compared to the smaller counterparts, which results in a peak EQE at lower peak current density. The fact that all the LEDs have the same maximum EQE at almost current density value, this is somewhat strange from the ABC model. This indicates there is something going on from the internal quantum efficiency aspect, but the authors did not address this

Response

Indeed, we should assure a similar Auger recombination rate for all the devices (or a size-

independent Auger recombination rate) to use current densities at the peak EQE to evaluate SRH nonradiative recombination, according to the ABC model. Olivier et al. (Appl. Phys. Lett. Vol. 111, 022104 (2017)) investigated experimentally Auger recombination coefficients of GaN blue LEDs with sizes ranging from $4 \times 4 \mu\text{m}^2$ to $500 \times 500 \mu\text{m}^2$. They found that the Auger recombination coefficient is size-independent for LEDs smaller than $100 \mu\text{m}$ (Fig. 5 of the above reference). The Auger recombination coefficient was indeed found to increase slightly when the chip size was increased to beyond $100 \mu\text{m}$. Therefore, it is reasonable to assume a size-independent Auger recombination rate for the micro-LEDs studied in this work.

In the present version of the revised manuscript, we added a few sentences to discuss the influence of Auger recombination rate and included the above paper as a new reference. Please see, line 178 – line 181, Ref. 22.

We hope the above revisions could answer your comments and we appreciate your consideration on the publication of our manuscript.

REVIEWERS' COMMENTS

Reviewer #2 (Remarks to the Author):

I have no further comments on the manuscript